



# LAND-SUITE V1.0: a suite of tools for statistically-based landslide susceptibility zonation

Mauro Rossi[1], Txomin Bornaetxea[2], Paola Reichenbach[1]

[1]Istituto di Ricerca per la Protezione Idrogeologica, Consiglio Nazionale delle Ricerche (CNR IRPI), Perugia, 06128, Italia

[2]Departamento de Geologia, Universidad del País Vasco / Euskal Herriko Unibertsitatea (UPV/EHU), Leioa, 48940, Spain

*Correspondence to*: Mauro Rossi (mauro.rossi@irpi.cnr.it)

**Abstract.** In the past 50 years, a large variety of statistically-based models and methods for landslide susceptibility mapping and zonation have been proposed in the literature. The methods, applicable to a large range of spatial scales, use a large variety

of input thematic data, different model combinations and several approaches to evaluate the models performance. Despite the numerous applications available in the literature, a standard approach for susceptibility modelling and zonation is still missing. The literature search revealed that several articles describe tools that apply physically based models for susceptibility zonation, but only few use statistically-based approaches. Among them, LAND-SE (LANDslide Susceptibility Evaluation) provides the possibility to perform and combine different statistical susceptibility models, and to evaluate their performances and associated

uncertainties. This paper describes the structure and the functionalities of LAND-SUITE, a suite of tools for statistically-based landslide susceptibility modelling which integrate, extend and complete LAND-SE. LAND-SUITE is able to: i) facilitate input data preparation; ii) perform preliminary and exploratory analysis of the available data; iii) test different combinations of variables and select the optimal thematic/explanatory set; iv) test different model types and their combinations; and v) evaluate the models performance and uncertainty. LAND-SUITE provides a tool that can assist the user to reduce some common source

of errors coming from the data preparatory phase, and to perform more easily, more flexible and more informed statistically-based landslide susceptibility applications.

## 1 Introduction

Landslide susceptibility measures the degree to which a terrain can be affected by future slope movements and provides an estimate of where landslides are likely to occur (Chacon et al., 2006; Guzzetti et al., 2005). A wide variety of statistically-

based models and methods for landslide susceptibility mapping and zonation have been proposed in the literature in the past 50 years. Statistically-based susceptibility models are applied to identify the functional (statistical) relationship between instability factors, described by sets of geo-environmental (independent) variables, and the known distribution of landslides, taken as the dependent model variable. This functional relationship is used to ascertain the propensity of the terrain to generate landslides, and to predict susceptibility.





A recent review article published by Reichenbach et al. (2018), has shown that more than 163 model type names are listed in the literature by different authors. The models were classified into 19 groups that allowed to highlight that logistic regression, neural networks, and data overlay model are the most used modelling approaches. The literature review also revealed a considerable variability of landslide and thematic data types, scales selected for the modelling and diversified choice of criteria used to evaluate the model performances. All these different issues, as well as their possible combinations, suggest that it is

possible to select and apply a vast and heterogeneous number of methodologies to assess landslide susceptibility. As a matter of fact, a standardized methodology, procedure and software for susceptibility assessment is still missing.

As an attempt to fulfill this gap, Reichenbach et al. (2018) in the final remarks suggest nine interrelated steps to prepare a reliable landslide susceptibility assessment and for the proper use of the associated terrain zonations (see Table 3 in Reichenbach et al., 2018). Such a methodological guideline allows for proceduralized but flexible susceptibility assessments,

although it assumes basic expertise and skills in geomorphology, data preparation, data analysis and geo-computation.

In the literature, are available several articles that describe tools suitable for the analysis of shallow landslides using physically based slope stability simulators (as for example, SHALSTAB, SINMAP, GEOtop-FS, HIRESSS, TRIGRS, r.slope.stability, etc), but very few articles propose software for statistically-based landslide susceptibility zonation. Among them, Brenning et al. (2008) provides an example of how GIS-based tools can be combined with powerful statistical models. Osna et al. (2014)

implemented GeoFIS, a tool developed with MATLAB, for the assessment of landslide susceptibility. GeoFIS includes two main open source libraries, one for GIS operations and the other for creating a Mamdani fuzzy inference system. Bragagnolo et al. (2020) developed r.landslide, a free and open source add-on to the open source GRASS software for landslide susceptibility mapping. The tool is written in Python language and works on the top of an Artificial Neural Network fed with environmental parameters and landslide databases. In 2020, Sahin et al. propose a tool package called Landslide Susceptibility

Mapping Tool Pack (LSM Tool Pack) for producing landslide susceptibility maps based on integrating R with ArcMap Software.

Rossi and Reichenbach (2016), following the previous experience described in Rossi et al. (2010), proposed LAND-SE (LANDslide Susceptibility Evaluation), a software designed to perform susceptibility modelling and zonation using different statistical models, combining ensemble of models and quantifying their performances and the associated uncertainties. The

software coded in R, is released with an open source licence and has the main intent to distribute a standard, widely accessible and repeatable tool to generate high-ranked quality landslide susceptibility zonation (Guzzetti et al., 2006), chiefly following the steps from 4 to 7 listed in Table 3 of  Reichenbach et al. (2018).

Despite this effort, the quality of the zonations produced with LAND-SE is still extremely variable, with the main sources of errors and uncertainty coming from the landslide susceptibility assessment preparatory phases (steps from 1 to 3 listed in Table

3 of Reichenbach et al., 2018). Indeed, a large complexity and a number of obstacles are present in these apparently basic but highly relevant steps for susceptibility evaluations.

To better support the overall landslide susceptibility assessment process, we have designed and implemented the LAND-SUITE software (LANDslide - SUsceptibility Inferential Tool Evaluator), which integrate, extend and complete LAND-SE.





The new code allows to: i) facilitate data preparation; ii) perform preliminary and exploratory analysis of the available data;
iii) test different combinations of variables and select the optimal thematic/explanatory variable set; iv) test different model types and their combinations, and v) evaluate the models performance and uncertainty. In synthesis, LAND-SUITE provides the user with the possibility to perform more easily, more flexible and more informed statistically-based landslide susceptibility applications and zonations.

The article illustrates the major functionalities offered by LAND-SUITE, including inputs and outputs. Section 2 describes the
main software data requirements and specifications. Section 3 describes the software modules and their functionalities, providing a basic background for their usage/interpretation; Section 4 illustrates the tool application in a test area, and Section 5 formalizes some final remarks. We have introduced a test area only with the purpose to show the most relevant results and outputs in a real application, but the critical analysis and discussion of the results are out of the scope of the article. The paper is completed by a supplement containing the software code and a user guide.

## 2 Data requirements and specifications


LAND-SUITE is a suite of R (R Core Team, 2021) tools aimed to support the landslide susceptibility inference process. It basically extends the LAND-SE software (Rossi and Reichenbach, 2016), which is mainly designed to perform statistically-based susceptibility modelling.

LAND-SUITE can use data in raster (GeoTiff) or vector (ESRI shapefile) format, depending on the selected type of application.
The input data can be provided with different resolutions (for rasters) and attributes (for vectors) and using different reference systems. The software can accept data stored in a local system folder, or alternatively, the data can be stored and imported from a GRASS GIS Mapset. The use of GRASS Mapset is preferred for large datasets because it facilitates the massive code execution via command line interface.

A prerequisite for the execution of LAND-SUITE, is the availability of two main input data: a landslide inventory map to be
used as a dependent variable in the susceptibility analysis, and a set of thematic maps to be used as independent explanatory variables. To execute the tools in raster mode, the user needs to provide all the inputs as raster layers, while in the vector mode the software requires a vector layer (e.g., a polygon-based slope unit partition layer) with the attribute table containing both the dependent (i.e. derived from the landslide inventory map) and the independent variables (i.e. the thematic information).

The selection and the preparation of model inputs is an important preliminary phase for the landslide susceptibility assessment
and for the execution of LAND-SUITE. The review by Reichenbach et al. (2018) revealed that different types of inventory maps (e.g., historical, geomorphological, event and multi-temporal landslide inventories) were chosen to derive landslide information used as a dependent variable in the modelling. Inventory data are used to classify the mapping units as 0 or 1, a largely accepted method to indicate the absence or presence of landslides respectively. This classification is also used in LAND-SUITE to describe the dependent or grouping variable.





Reichenbach et al. (2018) also highlighted that almost 600 different input thematic variables are mentioned in the literature as independent variables in the modelling. Such variables, which can be grouped in five main categories (i.e., geological, hydrological, land cover, morphological, and other types of variables), can be either continuous (e.g., slope, elevation, etc.) or categorical (e.g., lithology, land use, etc.). Depending on data types, thematic variables require different pre-processing phases to be properly used in the statistical models. In particular, the continuous variables can be used directly in the analyses

implemented in LAND-SUITE, while the categorical ones need to be converted into dummy variables. Dummy variables represent categorical data using distinct numerical categories and can be derived using the following approaches:

- Heuristical (based on personal judgment/experience), which is performed assigning to different categories an index value that increases/decreases with the expected propensity to instability (Carrara, 1983);
- Numerical, calculating the relative landslide incidence in each category (Lee and Min, 2001; Yilmaz, 2009; Trigila

et al., 2015; Van Westen et al., 1997);
- Numerical, calculating the percentage of each category in the mapping unit, generally adopted when using polygon-like terrain partitions (Carrara, 1983; Carrara et al., 2008).

Another common data pre-elaboration step in multivariate statistics is the variable scaling (Becker et al., 1988; Han et al., 2011). The data scaling may hold a strong statistical meaning, particularly in the case of normally distributed variables (i.e.

following a Gaussian distribution), transforming their distribution into a standard normal distribution (i.e., with mean value equal to 0 and standard deviation equal to 1). In all the other cases, such transformation chiefly aims at making variable values ranging on a similar scale, improving their comparability. In many cases, the scaling does not lead to significantly different susceptibility results, but in some conditions, it may avoid problems on numerical model convergence. LAND-SUITE includes a specific option to perform scaling that will be discussed later, but this preliminary transformation can be also done in a GIS

or other computational environments.

The maximum extension of the study area and the relative calculation times are strongly controlled by the data size and resolution and by the hardware characteristics, chiefly the RAM size and CPU speed. The code, which is essentially an R script, is executed in memory. During the execution and computations, the data are converted in a tabular format and stored at intermediate software execution steps, in the filesystem in the binary RDATA format.

During the software execution, LAND-SUITE provides outputs of specific analyses and evaluations in textual or graphical formats. At the end of the modelling computation, maps are also available as output in the classical GIS geographical formats.

### 3 Software description

LAND-SUITE is composed by three modules:

- LAND-SIP: LANDslide - Susceptibility Input Preparation;

- LAND-SVA: LANDslide - Susceptibility Variable Analysis;
- LAND-SE: LANDslide - Susceptibility Evaluation.





The three modules are coded as separate .R script files and can be executed under different operating systems.

The common LAND-SUITE run starts with LAND-SIP, which is able to execute in cascade LAND-SVA and successively LAND-SE. Alternatively, only one of these last two modules can be executed after LAND-SIP, depending on the user needs

and on the type of software applications. The three modules can also be executed separately, as long as the user is able to provide the appropriate data input.

### 3.1 LAND-SIP: LANDslide - Susceptibility Input Preparation

LAND-SIP is designed for the input preparation and has a high relevance for the susceptibility analysis, because its main purpose is the subdivision and preparation of the training and validation datasets, that will be used by the other two modules.

The dataset partition is controlled and customized by the user, that can select the type of the mapping unit (i.e., raster or polygons), choose the appropriate combination of variables, define the extent (i.e., using a mask) of the training and the validation areas, and choose the output types. This large number of options allows  the user to decide and perform largely diversified types of susceptibility applications. LAND-SIP allows the user to select different functionalities and criteria to partition the training and validation datasets:

- *Balanced or unbalanced random sampling*. In the balanced sampling, an equal number of mapping units with grouping values equal to 0 and 1 are selected randomly. Conversely, in the unbalanced sampling the proportions of mapping units with grouping values equal to 0 and 1 is different and is defined by the user. In the raster-based analyses, the user may choose two ways to select the mapping units with landslides: i) *pixels sampling*, based on a pixels' random sampling within mapped landslides, and ii) *landslides sampling*, based on a random landslides

sampling (using an additional landslide vector layer), where all the pixels of a selected landslide are considered either part of the training or of the validation datasets;

    - *Subsampling, or sampling reducing partitions*. In subsampling the size of the original dataset is randomly reduced by the user specifying the proportion of data used. This criterion is particularly helpful for preliminary investigations, in applications with large datasets or in case of limited computation resources;

- *Spatially or temporally-based datasets partition*. This criterion uses different input layers for the training and validation;

    - *Combinations of the criteria* described above.

The criteria are fully customizable by the user. Once a given criteria is chosen and training and validation datasets correctly partitioned, all the subsequent analyses will be performed accordingly. As previously mentioned, such datasets are always

stored in RDATA format to guarantee full data handling and control. Detailed information on LAND-SIP configurations can be found in the user guide.

The flexibility of the choices in the configuration phase allows the user to draw and execute many diversified susceptibility applications. It is out of the scope of this paper, if not impossible, to identify all the possible potential software applications. However, in the following, five applications (i.e, hereafter referred to as "Cases") are listed and discussed, with the purpose of





explaining how LAND-SIP, and in turn LAND-SUITE, can be configured and used for executing the most common susceptibility investigations (Figure 1).

**Case A**: The susceptibility modelling is performed applying a regular cross validation approach. A balanced random sampling is used to select the grouping variable mapping units following the "pixels sampling" selection criteria, with the size of training and validation datasets (e.g., 70% training and 30% validation) selected by the user (Figure 1 Case A). This configuration is

usually applied for exploratory analysis mainly focused on the preliminary evaluation of the explanatory variables (see LAND-SVA section), and of the statistical performance of the model. This execution can be performed by the user to select, add or remove explanatory variables before the application of the trained  model to the entire study area (Case C).

**Case B**: This application considers a cross validation approach similar to Case A, but the training and validation datasets partition uses the "landslides sampling" selection criteria. As before, a balanced random sampling and a specific size of the

training and validation datasets (e.g., 70% training and 30% validation) are chosen (Figure 1 Case B). As in Case A, it can be used to analyse the explanatory variables and to test the modelling results as well as its dependency from the selection of different landslide samples.

**Case C**: The training configuration can be similar either to Case A or B, but the validation is applied to the entire study area. This case should be applied when the definitive set of explanatory variables is selected and the statistical performance of the

model is satisfactory and acceptable. The validation map will show the susceptibility zonation for the entire extent of the study area (Figure 1 Case C).

**Case D**: This case performs a temporal validation, applicable when a geomorphological/historical inventory map is available to train the model and an event (or a successive) landslide inventory map is used for validation. In such a case the landslide event map used for the model validation may cover only a portion of the study area, with a spatial extent different from the

inventory map used for the calibration (Figure 1 Case D). This configuration requires two different mask files, one covering the entire study area and the other only the area affected by the event. The selection of the explanatory variables and the preliminary evaluation of the model can be performed applying Case A or B. The temporal validation may cover the entire study area, when an event inventory is available for its total extent.

**Case E**: This case performs a spatial validation, with the model calibration performed in a given region of the study area and

the validation in a different one. For example, the model training and validation can be performed in two contiguous but not overlapping river basins. In such a case, the variable selection and the preliminary model testing could be performed only in one of the two basins similarly to Case A or B. In this case the explanatory variables and landslide inventory map should be available in the two regions with the same characteristics (Figure 1 Case E).  This configuration requires two different landslide inventory maps, two mask files and two explanatory variables datasets, respectively for the calibration and for the validation

region.

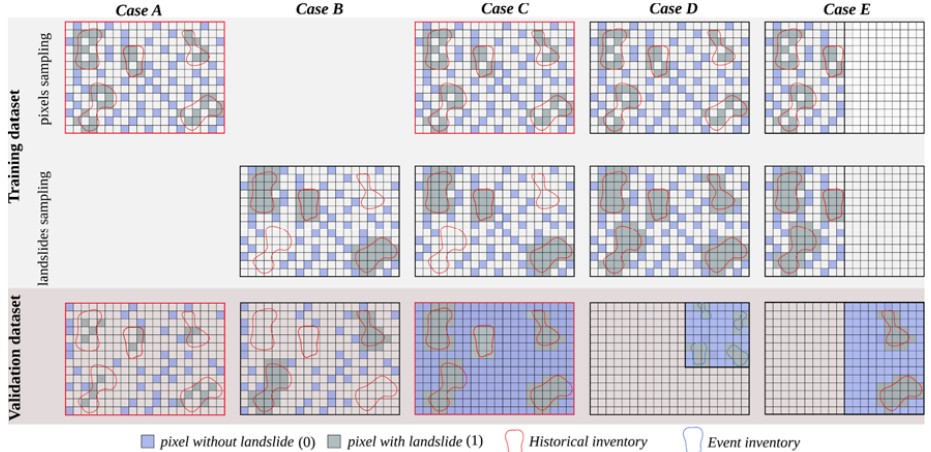

**Figure 1: Simplified representations of the five LAND-SUITE applications, referred to as "Cases" in the figures and text, representing common susceptibility investigations. Red boxes highlight the cases described in the application (Section 4).**

### 3.2 LAND-SVA: LANDslide - Susceptibility Variable Analysis

LAND-SVA is designed for the explorative analysis of the LAND-SE training and validation input datasets and facilitates the selection of the optimal set of variables. The tool automatically detects continuous or dummy variables (i.e., derived from categorical data and normally represented with numerical discrete values) and selects the outputs accordingly. All the analyses are performed separately for the training and validation datasets, with the main purpose to provide the possibility to analyse and control the dataset differences.

In this step, the user may decide whether or not to scale the variables and the option is applied jointly to the two datasets, to guarantee the comparability and applicability of the trained susceptibility model to the validation datasets. The variable scaling introduces advantages, particularly during numerical models convergence, avoiding working with variables with diversified ranges. However, two susceptibility analyses, performed with scaled or not scaled variables, lead to the same results when both are able to converge. It is important to note that two analyses performed using scaled variables in two different areas, do not necessarily guarantee the comparability of the variable coefficients. Similarly, such comparability does not hold for coefficients of variables derived at different data resolutions (e.g. coefficients of slope derived using two different DEM resolutions).

LAND-SVA performs the following analyses on continuous and categorical input variables (Figure 2 and 3, Table 1):

- **Conditional density analysis** (Figure 2):





- o **Density plots** for continuous variables that show the distribution of the values of numeric variables, stratified by the corresponding grouping variable value (0 and 1). Such plots use a kernel density estimator to show the probability density function of the variable. It basically corresponds to a smoothed version of a histogram plot and can be interpreted similarly;

- o **Conditional density plots** for continuous variables that examine the proportion of the grouping variable values (0 and 1) against the variation of a given continuous variable;

- o **Histogram plots** for categorical variables that, similarly to density plots, show the distribution of the values of categorical variables stratified by the corresponding grouping variable value (0 and 1). These plots use a normalized histogram counting to estimate the probability density function;

- o **Mosaic plots** for categorical variables that, similarly to conditional density plots, show the proportion of the grouping variable values (0 and 1) for different variable categories;

- **Pairwise correlation analysis** (Figure 3) of the input variables; in the analysis a correlogram chart and a correlation matrix are prepared to show pairwise correlation statistics among the different explanatory variables. The correlogram shows: in the upper triangular matrix, the values of the Pearson correlation coefficient for each pair of variables (i.e.; R coefficient ranging between −1 and 1, respectively for a perfect negative and positive correlation); in the lower triangular matrix, a graphical representation of the level of correlation (i.e.; flattened negatively and positively oriented ellipses, respectively for a negative and positive correlation); in the diagonal, the R value for the correlation of a variable with itself (R=1). Colours indicate different levels of correlation (i.e.; white for no correlation, red and blue respectively for negative and positive correlations);

- **Multicollinearity test** (Table 1) of the input variables; the analysis follows the diagnostic procedures described by Belsley et al. (1980), which examines the conditioning of the matrix of independent variables computing a test statistic called condition index. In LAND-SVA, a multicollinearity table is prepared to identify multicollinearity among the explanatory variables. Multicollinearity exists whenever a variable is highly correlated with one or more of the other variables and represents a problem undermining the statistical significance of the independent variables.

Multicollinearity implies that one variable in a multiple regression model can be linearly predicted from the others, with a substantial degree of accuracy.





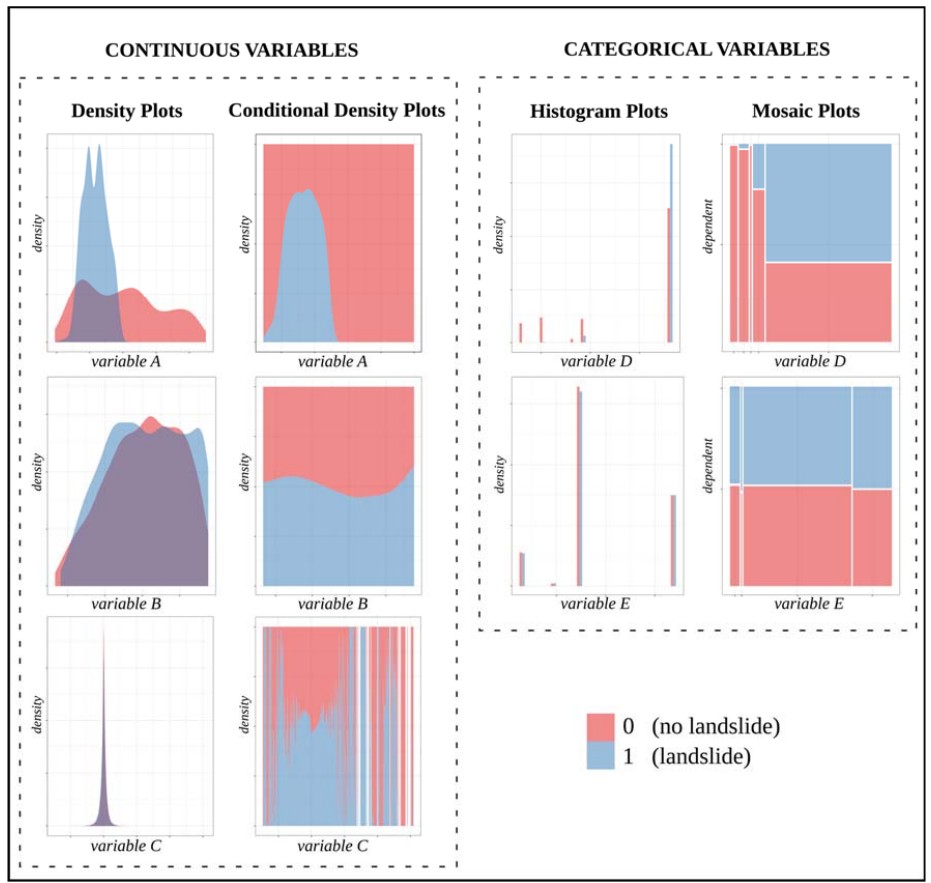

**Figure 2: Example of the conditional density analysis outputs generated by LAND-SVA for 5 synthetic explanatory variables.**

Some guidance is provided for the interpretation of the conditional density outputs shown in Figure 2. The density and histogram plots highlight significant numerical and categorical variables when the distribution of the values corresponding to the grouping variable categories (0 or 1) are significantly different (i.e., different shapes and lack of overlapping). Only under these circumstances, a variable may have a high significance in the modelling. The conditional density and mosaic plots need to be interpreted considering the variation and trend of the proportion of the grouping variable categories (i.e., the proportion

of 0 or 1 along the vertical axis) along with the variable value (i.e., along the horizontal axis). A distinct increase or decrease of such proportion, along with a reduced oscillation of it, and without lack of data, is the expected behaviour to identify a variable contributing significantly to the susceptibility zonation. Under these circumstances, an independent explanatory





variable may have an unambiguous effect on the dependent grouping variable used in the modelling (i.e., the presence or absence of landslides in the mapping unit). Following these considerations, only the variables A and D should be considered in the susceptibility modelling (Figure 2).

The pairwise correlation analysis and multicollinearity test are easier to interpret. When a significant high correlation is detected among two or more variables, one or more of the correlated variables should be excluded from the analysis. This is relevant for the following reasons:

- the joint use of two or more correlated variables does not introduce a significant advance for the multivariate modelling;
- generally, multivariate models assume independence among explanatory variables and when correlation exists, the independence assumption is not verified;
- when the degree of correlation among variables is high, it can introduce problems during the model fitting and for the interpretation of the model results;
- multicollinearity can introduce two main types of problems: i) the coefficient estimates can vary largely depending on the other independent variables considered in the model, with such coefficients' values becoming very sensitive to small model changes; ii) multicollinearity may reduce the precision of the estimated coefficients, weakening the statistical significance of the model, leading to a limited p-values reliability when identifying statistically significant independent variables;
- when collinearity occurs, the model coefficient values and their signs may change significantly depending on the specific variables included in the model, leading to difficulties to evaluate the results. Slightly different models may lead to different conclusions, making the actual contribution of variables impossible to understand.

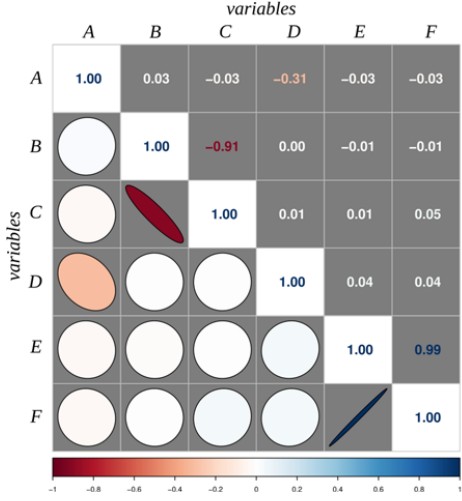





**Figure 3: Example of the graph showing the output of the pairwise correlation analysis, generated by LAND-SVA.**


| Condition index | Variable coefficients | | | | | | |
|---|---|---|---|---|---|---|---|
| | intercept | A | B | C | D | E | F |
| 1.000 | . | . | . | . | . | . | . |
| 1.031 | . | . | . | . | . | . | . |
| 1.246 | . | . | . | . | . | . | . |
| 1.421 | 0.999 | . | . | . | . | . | . |
| 1.711 | . | 0.654 | . | . | 0.654 | . | . |
| 4.555 | . | . | . | . | . | . | . |
| 746341312145290.375 | . | . | 1.0 | 1.0 | . | 1.0 | 1.0 |

**Table 1: Example of the table showing the output of the multicollinearity test generated by LAND-SVA.**

Pairwise correlated variables are those with a Pearsons' R values in the correlogram matrix close to +1 or -1 (Figure 3). Instead, multicollinearity is detected when the test statistic (i.e., the condition index in Table 1) is greater than 30 (Belsley, 1991).

When a large condition index (rows with condition index > 30) is associated with two or more variables with large variance decomposition proportions (values corresponding to variables > 0.5), these variables may cause collinearity problems. Based on the above considerations, the variables B, C, E and F show multicollinearity (Table 1) and the correlogram (Figure 3) helps to identify correlations between B and C, and E and F. These results suggest to exclude alternatively B or C (negatively correlated), and E or F (positively correlated).

**3.3 LAND-SE: LANDslide - Susceptibility Evaluation**

LAND-SE is the module for landslide susceptibility modelling and zonation that is described in detail in Rossi and Reichenbach (2016). The software holds on the possibility to perform and combine different statistical susceptibility modelling methods, evaluate the results and estimate the associated uncertainty. In particular, it allows for: i) the selection of different combinations of multivariate approaches; ii) the evaluation of the model prediction skills and performances using success

contingency matrices and plots, ROC curve and prediction rate curves; iii) the estimation of the associated uncertainty and errors; iv) the production of results in standard geographical formats (shapefiles, geotiff); and v) the usage of additional computational parameters to tune the calculation procedure for the analysis of large data sets.

The basic LAND-SE execution flow involves the following steps:

- the single susceptibility models' executions and zonation production;

- the combination of the single susceptibility models using a logistic regression approach;

- the evaluation of the single and combined susceptibility models;

- the estimation of the uncertainty of the single and combined susceptibility models.



Additional details on the LAND-SE tool specifications, configuration, functioning, and the scientific assumption can be found in Rossi and Reichenbach (2016), Rossi et al. (2010), as well as in the LAND-SUITE user guide.

**4 LAND-SUITE application**

To better illustrate the LAND-SUITE functionalities, we selected a study area in the Gipuzkoa Province, located in the northern sector of the Iberian Peninsula. In the area, a field-work based landslide inventory and 14 explanatory variables were used to investigate landslide susceptibility (Bornaetxea et al. 2018). In this article, the data are used to describe an application of LAND-SUITE, selecting for the purpose Case A and C described above. The application provides examples of the

susceptibility analysis outputs, including plots and maps. The critical discussion of results and their scientific relevance is out of the scope of this article and requires dedicated analysis, such as those described by Bornaetxea et al. (2018) and Rossi et al. (2021).

**4.1 Description of the study area and available data**

The Gipuzkoa Province is located in the northern part of the Iberian Peninsula, along the western end of the Pyrenees and

covers an area of 1980 km2, with an altitude ranging from the sea level to 1528 m a.s.l. The province, characterised by a steep morphology, is subdivided in six main watersheds that drain the territory toward the Cantabrian Sea (Figure 4). The investigated area is lithologically heterogeneous, with materials ranging from Paleozoic rocks to Quaternary deposits, corresponding to a hilly and mountainous Atlantic landscape (Mücher et al., 2010). The average annual precipitation is 1597 mm (González-Hidalgo et al., 2011) with two maximum rainy seasons: November–January and April.

The landslide inventory was prepared by an experienced geomorphologist during field surveys. The map shows the location and shape of 793 individual landslides in polygon format, mainly classified as shallow mass movements. A total of 14 geo-environmental maps were available as explanatory variables. Morphometric variables, such as elevation, slope, sinusoidal slope (Santacana Quintas, 2001; Amorim, 2012), aspect, surface area ratio (SAR), terrain wetness index (TWI), curvature, plan curvature and profile curvature, were derived from a DEM with a 5 m × 5 m spatial resolution. Lithology, permeability,

regolith thickness, land use and vegetation were downloaded from the official spatial data repository of the Basque Country (GeoEuskadi). Relative landslide incidence, by means of the Frequency Ratio (Bonham-Carter, 1994; Lee et at., 2002), was used to assign a numerical value to each category (hence transformed into dummy variables). For simplicity, we limited the model application to the two central and largest watersheds, which correspond to the Urola and Oria basins.

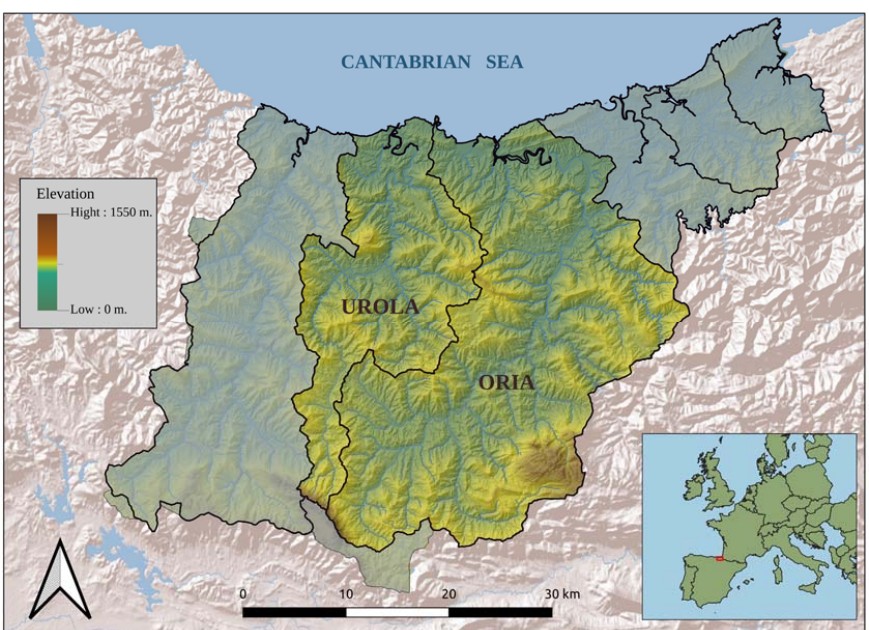

Figure 4: Location of the Gipuzkoa Province study area and the two river basins Urola and Oria.

**4.2 LAND-SIP: preparation of the training and validation datasets**

Among all the possible LAND-SUITE applications, we selected the cross-validation approach with pixel sampling method (Case A). Moreover, we applied the balanced random sampling criteria to select the same number of pixels with and without landslides, for both the training and validation datasets. The susceptibility model was calibrated using 70% of the data and validated using the remaining 30%.

As a first step, using LAND-SVA, we performed a preliminary evaluation of the available data. After the selection of the most significant explanatory variables, we evaluated the statistical performance of the calibrated model with the inspection of the susceptibility outputs produced by LAND-SE. At the final step, we applied Case C (Figure 1) to obtain a susceptibility zonation for the entire area.

**4.3 LAND-SVA: variables analysis and selection for the training and validation datasets**

We selected Case A and we ran LAND-SVA with the complete set of variables, for the explorative analysis of training and validation datasets in order to select the optimal combination of explanatory variables. The multicollinearity table (Table 2) shows one condition index larger than 30 and one close (29,722), with variance decomposition proportion values larger than





0.5. Thereby, the test detected two groups of variables (group I: curvature, planar curvature and profile curvature; group II:

SAR, slope, senoidal slope) with multicollinearity problems.

| Conditional index | Variable coefficients | | | | | | | | | | | | | | |
|---|---|---|---|---|---|---|---|---|---|---|---|---|---|---|---|
| | Int. | a | b | c | d | e | f | g | h | i | j | k | l | m | n |
| 1 | . | . | . | . | . | . | . | . | . | . | . | . | . | . | . |
| 1.117 | . | . | . | . | . | . | . | . | . | . | . | . | . | . | . |
| 1.521 | . | . | . | . | . | . | . | . | . | . | . | . | . | . | . |
| 1.583 | . | . | . | . | . | . | . | . | . | . | . | . | . | . | . |
| 1.804 | . | 0.699 | . | . | . | . | . | . | . | . | . | . | . | . | . |
| 1.841 | 0.997 | . | . | . | . | . | . | . | . | . | . | . | . | . | . |
| 1.949 | . | . | . | . | . | . | . | . | . | . | 0.537 | . | . | . | . |
| 2.078 | . | . | . | . | . | 0.695 | . | . | . | . | . | . | . | . | . |
| 2.156 | . | . | . | . | . | . | . | . | . | . | . | . | . | . | 0.644 |
| 2.453 | . | . | . | 0.573 | 0.669 | . | . | . | . | . | . | . | . | . | . |
| 2.799 | . | . | 0.758 | . | . | . | 0.692 | . | . | . | . | . | . | . | . |
| 3.001 | . | . | . | . | . | . | . | . | . | . | . | . | . | . | . |
| 3.637 | . | . | . | . | . | . | . | . | . | . | . | . | . | . | . |
| 29.722 | . | . | . | . | . | . | . | . | . | . | . | 0.892 | 0.998 | 0.981 | . |
| 323243.074 | . | . | . | . | . | . | . | 1 | 1 | 1 | . | . | . | . | . |

**Table 2. Multicollinearity table generated by LAND-SVA for the Gipuzkoa study area. Int: intercept, a: Aspect, b: Land-Use, c: Lithology, d: Permeability, e: Regolith thickness, f: Vegetation, g: Curvature, h: Planar Curvature, i: Profile Curvature, j: Elevation, k: SAR, l: Slope, m: Senoidal Slope, n: TWI.**





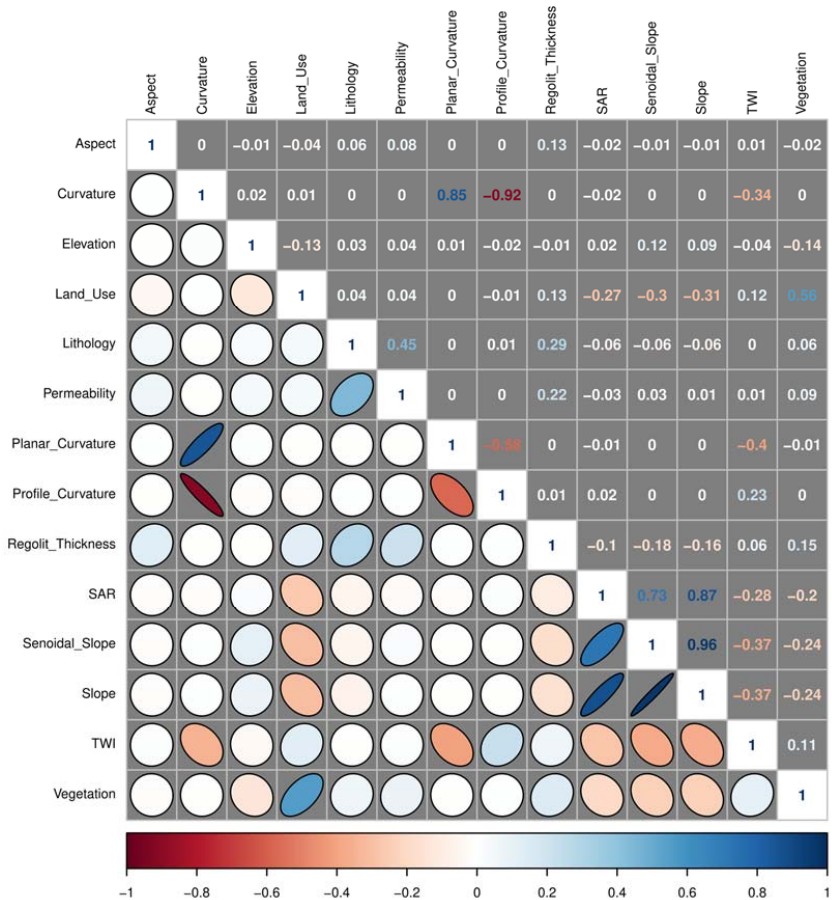


**Figure 5: The figure shows the correlogram obtained by LAND-SVA for the complete set of variables available in the Gipuzkoa study area (Case A).**

Inspection of the correlogram (Figure 5) confirms the pairwise correlations within groups I and II and highlights an additional

correlation between vegetation and land use, assuming a Pearsons' R absolute value of 0.5 as thresholds for detecting correlations.

To obtain additional information on the highly correlated continuous variables, the relation of each explanatory layer with the dependent variable was analyzed through the density plots and conditional density plots reported in Figure 6. Similarly, we




checked the histogram plots and mosaic plots (Figure 7) to analyze the categorical variables. All the remaining outputs of the
conditional density analysis were also evaluated, to check their relevance for the susceptibility modelling.

The evaluation of LAND-SVA outputs allowed:

- the removal of all the variables in group I, due to high correlation and to the lack of relevant differences between 1 and 0 in the density plots and conditional plots;
- the selection of slope in group II, based on the better distribution separation and trend shown in the density and
conditional plots;
- the selection of both vegetation and land use, with a weak correlation, confirmed by their Pearsons' R values only slightly higher than 0.5.



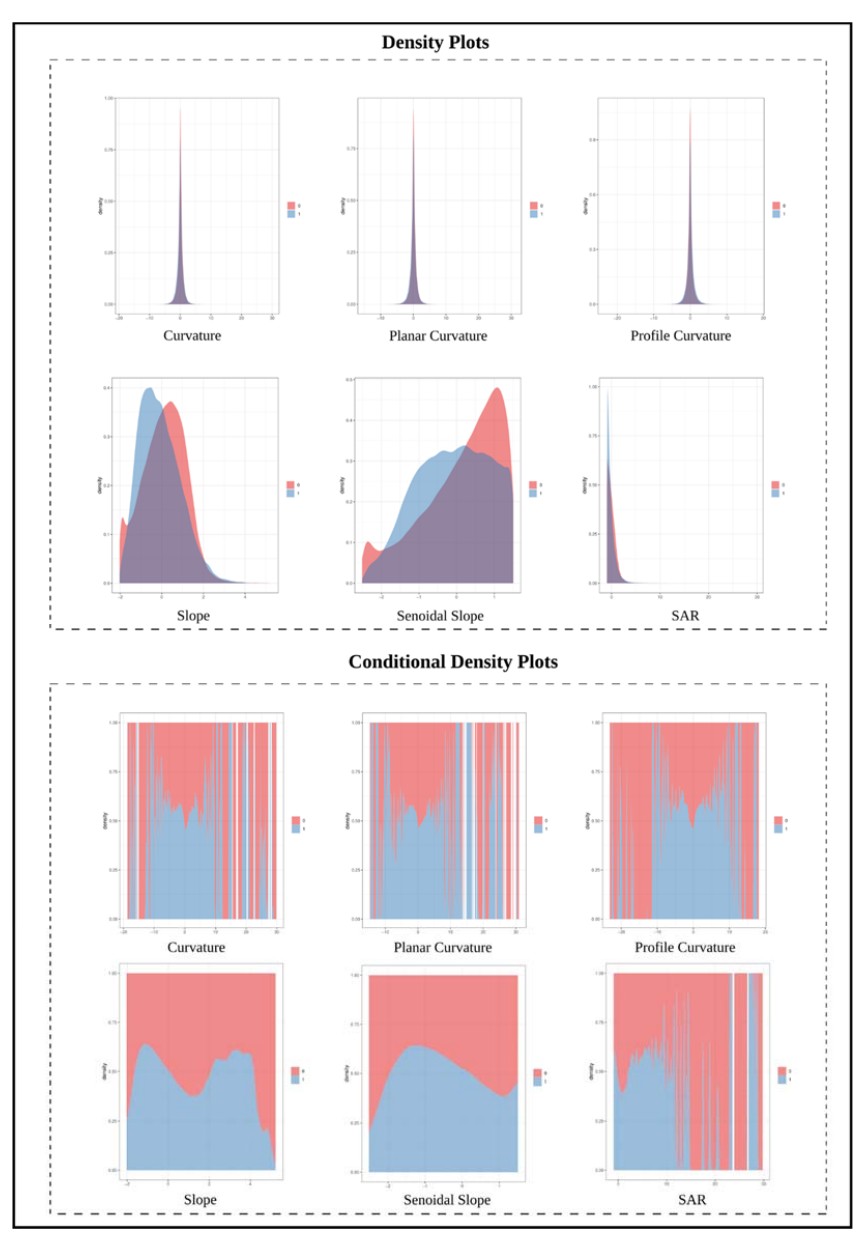





**Figure 6: Density plots and Conditional density plots for some continuous explanatory variables that show a significant correlation between them.**

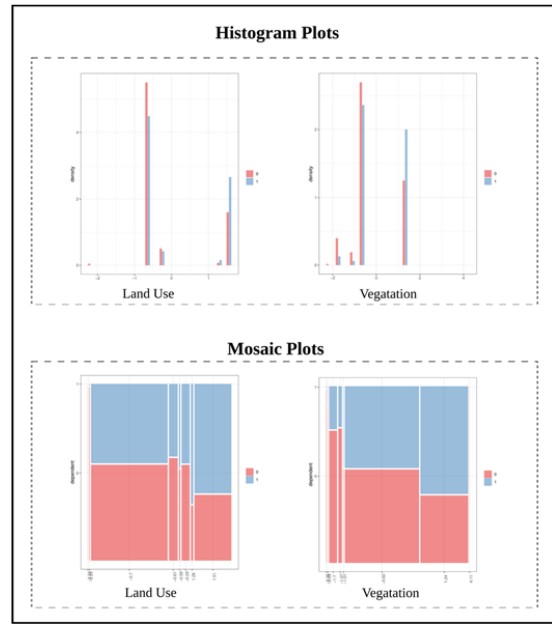

**Figure 7: Histogram plots and mosaic plots for two categorical explanatory variables that show a significant correlation.**


### 4.4 LAND-SE: susceptibility models' execution and zonation production

After the analysis of the results produced by LAND-SVA, the final set of explanatory variables used to run LAND-SE included aspect, land use, lithology, permeability, regolith thickness, vegetation, elevation, slope and topographic wetness index. The same training set was used to prepare the four single landslide susceptibility models and the combined model (Figure 8). The
figure shows the different landslide zonation maps and the plots (i.e., ROC plot, evaluation plot, success rate plot and contingency or fourfold plot) used to evaluate the training performance of the combined model. The two small maps at the bottom, illustrate the errors and uncertainty values associated with the combined susceptibility model (Rossi et al, 2010). This set of outputs, restituted by LAND-SE, is commonly used for the verification and analysis of the susceptibility zonations obtained by LAND-SUITE.





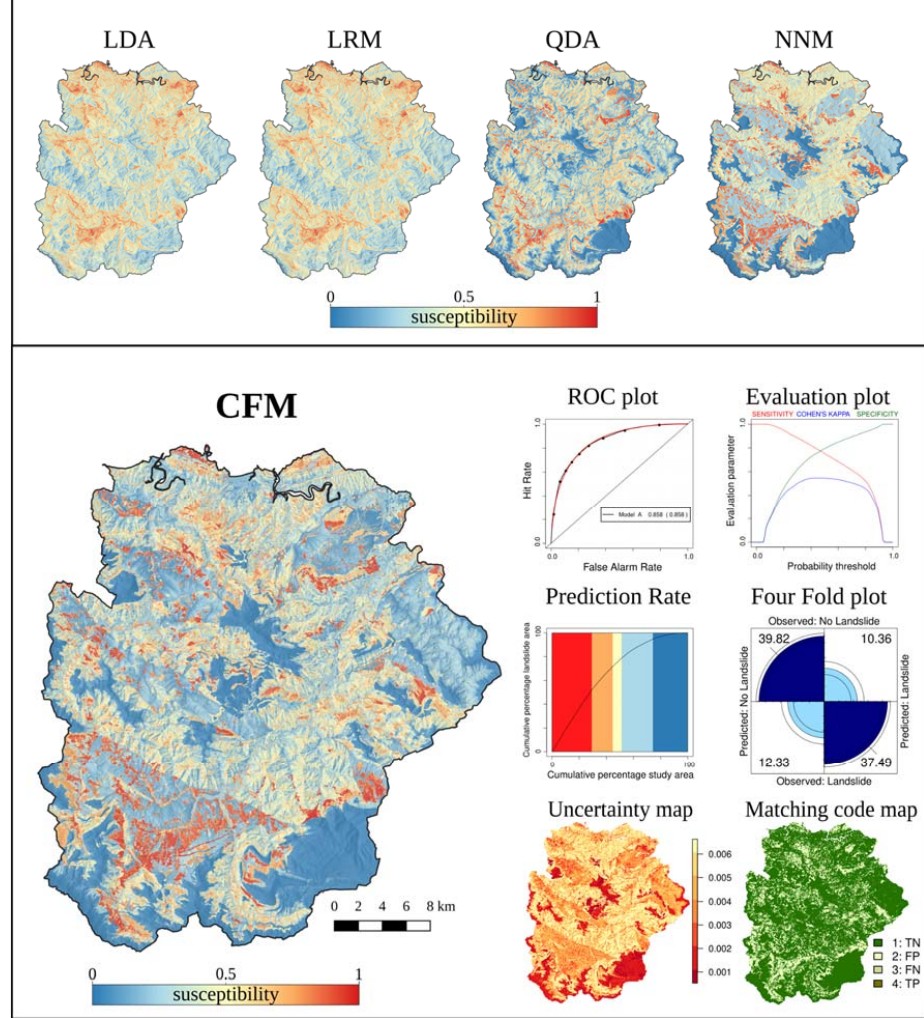


**Figure 8: Examples of most relevant outputs of LAND-SE obtained in the Gipuzkoa study area. LDA = Linear discriminant analysis; LRM = Logistic regression model; QDA = Quadratic discriminant analysis; NNM = Neural network analysis; CFM = Combined models' function.**





## 5 Final remarks

LAND-SUITE is a suite of tools for statistically-based landslide susceptibility zonation implemented in R and released with an open source license. The tool was developed to support the landslide susceptibility inference process, which is a complex task. LAND-SUITE was not designed to substitute the geomorphological/geological experience and competence of the operator, but to facilitate the preparation and the selection of the variables/data required for a reliable statistical analysis.

LAND-SUITE can be used to test diversified geomorphological hypotheses allowing to verify and discuss initial assumptions and to prepare less subjective statistically-based susceptibility zonation. Few initial skills are required to run the software, but the learning curve is not really steep. A user with a good knowledge of LAND-SUITE is able to run different models changing options, configurations and data.

   A key step for a reliable landslide susceptibility modelling, is the preparation of robust and unbiased input data, which largely 390 depends on the user's skill and experience. In many cases, the data classification approaches, reliability and representativeness of the thematic information are more important than the statistical methods and tools used for the landslide susceptibility estimation. Low quality output and errors derive often from incomplete or not significant data. The tool has the ambition to help the preparation of statistically correct and robust models, allowing to apply classical and standard statistical procedures (e.g., random sampling, data scaling, use of common machine learning approaches and standard evaluation metrics).

Moreover, LAND-SUITE can be applied using different mapping units (e.g., pixel, slope units, administrative units, etc.), with distinct configurations and data resolution at diverse spatial scales. The tool uses standard geographical formats in input and output and can facilitate the massive code execution via command line interface.

   The suite has high flexibility and allows to perform different partitions of the training/validation dataset and diversified validation tests (e.g., temporal, spatial, cross validation, etc.).

LAND-SUITE can be used to model and evaluate the spatial probability of the occurrence of other types of natural phenomena, such as floods, forest fires, rock falls source areas (Rossi et al., 2021). Indeed, as an open source tool, LAND-SUITE can be easily modified by a R programming skilled user and adapted to any specific needs.

## 6 Code availability and licence

   LAND-SUITE is composed of three modules (LAND-SIP, LAND-SVA, LAND-SE) coded as separate .R script files and can 405 be executed under different operating systems. The software was mainly tested under WindowsOS and LinuxOS, with the version of R-4.1.1 (64bit). Some code functionalities of LAND-SIP require GRASS GIS binding. We tested the script using GRASS GIS version 7 under WindowsOS and LinuxOS. We recommend LinuxOS, due to the better software integration at a bash scripting level.

   LAND-SUITE is free software; it can be redistributed or modified under the terms of the GNU General Public (either version 410 2 of the license, or any later version) as published by the Free Software Foundation. The program is distributed in the hope





LAND-SUITE V1.0 is archived in ZENODO repository with the DOI: 10.5281/zenodo.5650810.

## 7 Data availability

In this work, example data have been used only to show different LAND-SUITE applications and they are not needed to apply LAND-SUITE elsewhere. The software can in fact be used in other areas using the appropriate input data.

## 8 Author contribution

MR conceptualized the work, designed the overall methodology behind the software and supervised the research activity; MR wrote the core of the codes LAND-SIP, LAND-SVA and LAND-SE; TB implemented specific functionalities of LAND-SIP

and LAND-SVA, reviewed the codes and performed the overall LAND-SUITE code validation/testing; PR participated to the LAND-SUITE code validation/testing; MR wrote the original draft of the manuscript; TB and PR largely contributed to the review, edit and writing of the manuscript.

## 9 Acknowledgments

The implementation and improvement of LAND-SUITE with respect to the version published by Rossi and Reichenbach

(2016) was funded using mainly internal funds. Txomin Bornaetxea was financially supported by the postdoctoral fellowship program of the Basque Government (grant numbers POS_2020_2_0010) in the framework of a scientific collaboration with the Geological Survey of Canada and during the scientific collaborations with the Geomorphological Group of the Research Institute for the Geo-Hydrological Protection in Perugia, Italian National Research Council (CNR-IRPI).

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
