# Peer review of "LAND-SUITE V1.0: a suite of tools for statistically-based landslide susceptibility zonation"

_Geoscientific Model Development, 2021_

## Author Comment (AC1)

**REPLY TO GMD-2021-343 REVIEWERS'COMMENTS**

In the following, we reply to the reviewer's comments and suggestions. The original comments/suggestions are in black, and our answers are in red. The added or modified text is in red italic.

**\*\*\*\*\*\*\*\*\*\*\*\*\*\*\*\*\*\*\*\*\*\*\*\*\*\*\*\*\*\*\*\*\*\*\*\*\*\*\*\*\*\*\*\*\*\*\*\*\*\*\*\*\*\*\*\*\*\*\*\*\*\*\***

**REPLY TO 'COMMENT ON GMD-2021-343'**

**ANONYMOUS REFEREE #1, 04 MARCH 2022**

Landslide susceptibility analysis predicts where landslides are likely to occur, which is very important for human. There have been many methodologies to assess landslide susceptibility. However, a standardized methodology, procedure and software for susceptibility assessment is still required. This study developed suite of tools for statistically-based landslide susceptibility modelling. However, the current version looks like a specification and the scientific contributions need to be stressed. In addition, how does this software provide a standardized susceptibility analysis or how can people use it at a standard way?

LAND-SUITE does not want to propose a standard procedure for landslide susceptibility assessment, but it is an attempt (i) to facilitate the preparation and the selection of the variables/data required for a statistical susceptibility modelling, (ii) to provide largely used statistical approaches to derive susceptibility zonations, and (iii) to give largely recognised metrics to evaluate such modelling outputs.

We carefully checked the use of the term "standard" in the manuscript and, where needed, we removed or substituted it with more correct terms throughout the text.

The differences between this study and Bornaetxea et al. (2018) should be well presented.

We thank the reviewer for the comment and we realized the text was unclear. This manuscript cites Bornaetxea et al. (2018) in section 4 "LAND-SUITE application" only as a reference for a detailed description of the study area and the available thematic data. To better clarify this, we modified the paragraph at the beginning of section 4, as follows:

*"To better illustrate the LAND-SUITE functionalities, we selected a portion of the study area located in the Gipuzkoa Province (northern sector of the Iberian Peninsula) where a landslide inventory and 14 explanatory variables were mapped (Bornaetxea et al., 2018). This set of thematic data is used to describe different applications of LAND-SUITE (i.e. Case A and C in Figure 1) and to provide examples of the susceptibility analysis outputs, including plots and maps."*

The authors stated that LAND-SUITE provides a tool that can assist the user to reduce some common source of errors coming from the data preparatory phase, and to perform more easily, more flexible and more informed statistically based landslide susceptibility applications. But I cannot get such information. How can this tool help to reduce the errors in the data preparation

and why is this tool more informed?

We thank the reviewer for the comment, and to avoid misleading interpretations, we simplified the sentence as follows:

*"LAND-SUITE provides a tool to assist the user during the data preparatory phase and to perform diversified statistically-based landslide susceptibility applications."*

Major concerns:

Abstract:

1. Line 10: The authors stated that some physically based models are available. What is the purpose of this statements?

The purpose of this statement is to introduce and stress the novelty of the work which focuses on software for statistically-based landslide susceptibility zonation. Indeed, while there are many physically-based tools for distributed landslide modelling, only a few exist for statistically-based modelling. To make this more evident we rephrased the sentence as follows:

"*The literature search revealed that several software and tools are available to evaluate regional slope stability using physically-based analysis, but only a few use statistically-based approaches.*"

2. What is the limitations of LAND-SE? What is the difference between LAND-SUITE and LAND-SE?

The text of the abstract (line 8 to 12) was modified as follows:

"*This paper describes the structure and the functionalities of LAND-SUITE, a suite of tools for statistically-based landslide susceptibility modelling which integrates LAND-SE. LAND-SUITE completes and extends LAND-SE, adding functionalities to i) facilitate input data preparation; ii) perform preliminary and exploratory analysis of the available data; iii) test different combinations of variables and select the optimal thematic/explanatory set.*"

3. Some results and conclusions in the specific applications should be added in the abstract.

As explained in Chapter 4 (line 302-303) the specific application is described only as an example of the software usage and the critical discussion of results and their scientific relevance is out of the scope of this manuscript and then we believe this should not be described in the abstract.

Introduction:

4. Line 59: shows why the authors further developed the LAND-SUITE model? If so, more details should be given rather than only cite the table in the existing literature.

Thanks for the comment. We recognize the citation was misleading and we removed the citation of the table.

5. Line 64: the advantage of LAND-SUITE over LAND-SE should be stressed.

We changed the text as follows:

*"To better support the overall landslide susceptibility assessment process, we have designed and implemented the LAND-SUITE software (LANDslide - SUsceptibility Inferential Tool Evaluator), which integrates LAND-SE, able to execute different susceptibility model types and to evaluate their performance and uncertainty. LAND-SUITE completes and extends LAND-SE, adding functionalities to i) facilitate input data preparation; ii) perform preliminary and exploratory analysis of the available data; iii) test different combinations of variables and select the optimal thematic/explanatory set."*

Data requirements and specifications

6. This section needs to be reorganized. Currently, it is not logical and difficult to get the central information.

We merged the chapter with the "Software description" modifying the text. Most of the text was moved in the User Guide.

Software description

7. A simple schematics can be provided to show the link of three modules.
8. A table can be added to show the functions of each module.

As suggested by the reviewer, we added a new figure (Figure 1). The other figures were renumbered accordingly.

LAND-SUITE application

9. The authors need to present how to use this tool at a standard way in the applications rather than just show some results.

The details of the use of LAND-SUITE are already described in the User Guide available as a Supplementary Material.

Final remarks

10. Some prospects and limitations can be provided in this section.

Some of the prospects and limitations are already briefly described in the "Final remarks" section, where we added the following text:

*"We acknowledge that LAND-SUITE does not consider all the statistical approaches for landslide susceptibility modelling and zonation, which can be potentially included in future software upgrades. Possible LAND-SUITE advancements can also be achieved by implementing new procedures to evaluate the variables' significance across the different statistical approaches."*

In addition, we modified the last paragraph as follows:

*"LAND-SUITE can be used to model and evaluate the spatial probability of the occurrence of other types of natural phenomena (such as floods, forest fires, rock falls source areas, e.g. see Rossi et al., 2021) and this use may highlight the need for specific code modifications and refinements. Indeed, as an open source tool, LAND-SUITE can be easily modified by a R programming skilled user and adapted to any specific needs."*

Code availability

    11. I cannot access the software via the doi: 10.5281/zenodo.5650810.

We checked the link, which correctly refers to the software repository. As for scientific articles, the DOI link can be resolved using specific web services like https://dx.doi.org/. To visualize/download the software in the repository, it is mandatory to request access (that can be done using the specific button). The link works correctly and we already got several access requests.

Minor concerns:

Line 25-30: add some references

We added new references

Line 41: add some references

We added the references to the software codes

Line 41: "are available"?

We modified the sentence

Line 49: propose -> proposed

Done

Line 55, give the version of R

This information is reported in chapter 6 "Code availability and licence"

---

## Author Comment (AC2)

**REPLY TO GMD-2021-343 REVIEWERS'COMMENTS**

In the following, we reply to the reviewer's comments and suggestions. The original comments/suggestions are in black, and our answers are in red. The added or modified text is in red italic.

**\*\*\*\*\*\*\*\*\*\*\*\*\*\*\*\*\*\*\*\*\*\*\*\*\*\*\*\*\*\*\*\*\*\*\*\*\*\*\*\*\*\*\*\*\*\*\*\*\*\*\*\*\*\*\*\*\*\*\*\*\*\*\*\***

**REPLY TO 'COMMENT ON GMD-2021-343'**

**ANONYMOUS REFEREE #2, 12 APRIL 2022**

This paper is written in detail and provides a practical tool including data preprocessing, variable analysis and landslide susceptibility zonation for people who engaged in landslide susceptibility assessment. However, this paper is more like the software instruction manual which mainly introduced the extension of LAND-SE. In general, there are few studies on model innovation, whether physical model or empirical statistical model development in this paper.

We thank the referee for the comments and suggestions that help us to improve the manuscript.

This software suite LAND-SUITE mainly based on statistically-based landslide susceptibility assessment models. As the prediction from statistically-based models is not only influenced by input variables, training sample distributions, but also the number and representative of samples, how to carry out landslide susceptibility assessment under the condition of lack of samples is also one of the problems that need to be solved.

The manuscript describes LAND-SUITE and its functionalities. The set of tools implemented in LAND-SUITE is helpful to investigate different issues related to the training and validation datasets, their characteristics, limitations and representativeness. Indeed, to test issues highlighted by the reviewer, there are many possible approaches that can be used when selecting, generating and partitioning training and validation datasets. As we mentioned in the manuscript, the tool facilitates these types of analyses, but it doesn't want to substitute the user's expertise and knowledge, which should be investigated with specific analyses.

I would like suggest the authors further discuss the physically based models in this paper, or consider physical model as one of the directions of software extension in future research.

We think LAND-SUITE should be only focused on statistically-based landslide susceptibility zonation. In the literature, there are already articles that describe tools suitable for the analysis of landslides using physically-based slope stability tools (as for example, SHALSTAB, SINMAP, GEOtop-FS, HIRESSS, TRIGRS, r.slope.stability, etc). Those are already listed in the original manuscript from line 41 to line 43.

This also goes in the direction of many scientific contributions in susceptibility modelling, which are pushing towards the extensive use of AI (Artificial Intelligence) and ML (Machine Learning) approaches, which are proving to be effective and reliable in many applications. We do not believe that LAND-SUITE should go in the direction proposed by the reviewer, which is certainly interesting but out of the scope of LAND-SUITE.

Line 35-36, "As a matter of fact, a standardized methodology, procedure and software for susceptibility assessment is still missing." How to define the standardized methodology for landslide susceptibility mapping?

LAND-SUITE does not want to propose a standard procedure for landslide susceptibility assessment, but it is an attempt (i) to facilitate the preparation and the selection of the variables/data required for a statistical susceptibility modelling, (ii) to provide largely used statistical approaches to derive susceptibility zonations, and (iii) to give largely recognised metrics to evaluate such modeling outputs.

We carefully checked the use of the term "standard" in the manuscript and where needed we removed or substituted it with more correct terms throughout the text.

Line 41, "In the literature, are available….". Grammar mistake.

We modified the text as follows:

"*In the literature, several articles describe tools suitable for the analysis of shallow landslides using physically based slope stability simulators (as for example, SHALSTAB by Dietrich & Montgomery, 1998;, SINMAP by Pack et al., 1988;, GEOtop-FS by Simoni at al., 2008;, HIRESSS by Rossi et al.; 2013, TRIGRS by Baum et al., 2008, r.slope.stability by Mergili et al., 2014, etc), but very few articles propose software for statistically-based landslide susceptibility zonation.*"

In Fig. 8, it is clear that landslide susceptibility map from LDA and LRM is quite similar, while the spatial pattern of landslide susceptibility map from QDA and NNM is similar in most places. The NNM result mainly contribute to the final CFM result. Please further explain the possible reasons.

As we mentioned at the beginning of chapter 4, the critical discussion of results and their scientific relevance is out of the scope of this article and requires dedicated analysis, such as those described by Bornaetxea et al. (2018) and Rossi et al. (2021). In addition, the aspects related to the similarities/differences among the different susceptibility maps obtained from the different classification models have been already discussed. For example see the comparison of the spatial pattern and susceptibility values provided in Figure 8 of the cited article Rossi et al, 2010 (Rossi M., Guzzetti F., Reichenbach P., Mondini A., Peruccacci S. (2010) Optimal landslide susceptibility zonation based on multiple forecasts. Geomorphology, Vol. 114, 129-142, doi:10.1016/j.geomorph.2009.06.020).

---

## Author Response (AR2)

**REPLY TO GMD-2021-343 REVIEWER COMMENTS**

In the following, we reply to the editor and the reviewer comments. The original comments are in black, and our answers are in red. The added or modified text is in red italic.

**\*\*\*\*\*\*\*\*\*\*\*\*\*\*\*\*\*\*\*\*\*\*\*\*\*\*\*\*\*\*\*\*\*\*\*\*\*\*\*\*\*\*\*\*\*\*\*\*\*\*\*\*\*\*\*\*\*\***

**REPLY TO COMMENTS ON GMD-2021-343'**

**EDITOR, 22 MAY 2022**

As one of the reviewers pointed out, this version still looks like a software specification. And the scientific contributions need to be emphasized. Please restructure the manuscript and highlight the scientific contributions (not functionality contributions) after LAND-SE.

See the replies below.

**\*\*\*\*\*\*\*\*\*\*\*\*\*\*\*\*\*\*\*\*\*\*\*\*\*\*\*\*\*\*\*\*\*\*\*\*\*\*\*\*\*\*\*\*\*\*\*\*\*\*\*\*\*\*\*\*\*\***

**REPLY TO 'COMMENT ON GMD-2021-343'**

**ANONYMOUS REFEREE #1, 21 MAY 2022**

The current version still looks like a software specification. However, the scientific contributions of this manuscript are unclear and still need to be stressed.

Thanks for the comment. Unfortunately, we do not completely understand what is unclear and what needs to be stressed throughout the entire manuscript. The article is composed of two documents: the text and the software specifications. We have explained in text what we assume to be the scientific contribution of the new version of the tool and in the manual all the software specifications.

In chapter 3.2 we have described different analyses on continuous and categorical input variables. We consider Chapter 4 as an example of the scientific contribution of the tool to the improvement of landslide susceptibility assessment. The use of LAND-SUITE can optimize the selection and the combination of the variables that are an essential and significant component for landslide susceptibility assessment. In addition, LAND-SUITE can be used to test diversified geomorphological hypotheses allowing to verify and discuss initial assumptions and to prepare less subjective statistically-based susceptibility zonation.

To highlight more clearly the scientific contributions of the manuscript, we have largely modified the "Final remarks" section and we decided to rename this section in "Scientific contribution and final remarks" to better explain its content. Please find the modified text of the section in the following.

*5 Scientific contributions and final remarks*

*LAND-SUITE was developed to support the landslide susceptibility inference process, which is a complex task. LAND-SUITE includes a suite of tools for statistically-based landslide susceptibility zonation implemented in R and released with an open source license.*

*As highlighted by Reichenbach et. (2018), only a reduced number of scientific contributions on statistical landslide susceptibility modelling, properly select and combine the suitable variables and apply the relevant statistical evaluations for realising high-quality zonations. This is mainly due to the lack of a comprehensive and shared approach for susceptibility modelling. LAND-SUITE can be used for the preparation and the selection of the variables/data required for a reliable statistical analysis and it is designed to support the geomorphological/geological experience and competence of the operator. . LAND-SUITE facilitates and simplifies the testing of diversified geomorphological hypotheses allowing the verification and discussion of the initial modelling assumptions, the preparation of less subjective statistically-based susceptibility zonation and the evaluation of the quality of the modelling results. A key step for a reliable landslide susceptibility modelling, is the preparation of robust and unbiased input data, which largely depends on the user's skill and experience. In many cases, the data classification approaches, reliability and representativeness of the thematic information are more important than the statistical methods and tools used for the landslide susceptibility estimation. Low quality output and errors often derive from incomplete or not significant data. The tool has the ambition to help a skilled user with the preparation of statistically correct and robust models, allowing to apply and test easily different classical statistical procedures (e.g., random sampling, data scaling, use of common machine learning approaches and commonly-used evaluation metrics).*

*Using LAND-SUITE, the user can compare results of different mapping units (e.g., pixel, slope units, administrative units, etc.), with distinct configurations and data resolution at diverse spatial scales. LAND-SUITE does not consider all the statistical approaches for landslide susceptibility modelling and zonation, which can be potentially included in future software upgrades. Possible LAND-SUITE advancements can also be achieved by implementing new procedures to evaluate the variables' significance across the different statistical approaches.*

*The suite has high flexibility and allows to perform different partitions of the training/validation dataset and diversified validation tests (e.g., temporal, spatial, cross validation, etc.), which are relevant evaluation steps to realise robust scientific susceptibility modelling exercises.*

*LAND-SUITE can be used to model and evaluate the spatial probability of the occurrence of other types of natural phenomena (such as floods, forest fires, rock falls source areas, e.g. see Rossi et al., 2021) and this use may highlight the need for specific code modifications and refinements.*